# Multilevel Laser Induced Continuum Structure

**DOI:** 10.3390/e23070891

**Published:** 2021-07-13

**Authors:** Kaloyan Zlatanov, Nikolay Vitanov

**Affiliations:** 1Department of Physics, Sofia University, James Bourchier 5 Blvd, 1164 Sofia, Bulgaria; vitanov@phys.uni-sofia.bg; 2Institute of Solid State Physics, Bulgarian Academy of Sciences, Tsarigradsko Chaussée 72, 1784 Sofia, Bulgaria

**Keywords:** LICS, ionization, continuum, Fano profile, multilevel, trapping condition, population trapping

## Abstract

Laser-induced-continuum-structure (LICS) allows for coherent control techniques to be applied in a Raman type system with an intermediate continuum state. The standard LICS problem involves two bound states coupled to one or more continua. In this paper, we discuss the simplest non-trivial multistate generalization of LICS which couples two bound levels, each composed of two degenerate states through a common continuum state. We reduce the complexity of the system by switching to a rotated basis of the bound states, in which different sub-systems of lower dimension evolve independently. We derive the trapping condition and explore the dynamics of the sub-systems under different initial conditions.

## 1. Introduction

Coherent manipulation of quantum states is at the core of contemporary quantum physics. The development of analytical models [1,2,3,4,5,6] treating laser interaction between two, three, and more levels [7,8] is fundamental for the understanding and the demonstration of effects like rapid adiabatic passage (RAP) [9], stimulated Raman adiabatic passage (STIRAP) [10], and electromagnetically induced transparency (EIT) [11,12,13], to name just a few. Generally, the problems of coherent population transfer and state preparation involve only bound states that lie deep inside the potential of the atom, which have well defined discrete energies. There exist, however, unbound states distributed continuously in energy, and therefore termed continuum states to which the system can be coupled, for example, by the interaction with a strong laser by single or multi-photon absorption.

The simplest, flat continuum possesses no structures. However, structures in the continuum can be induced by laser fields. The emergence of resonance structures in the continuum has been described by Fano in his seminal paper [14] on autoionization. If a bound state is coupled to the continuum by a strong laser field, the latter “embeds” this state into the continuum. Scanning through this energy range by another (weak) laser field which couples a second state to the same continuum reveals a resonance structure known as *laser-induced-continuum-structure (LICS)* [15,16]. A suitable choice of the two-photon detuning between the two bound states, given by the *trapping condition* [17], allows for suppressing the ionization in theory to zero. In experiments, ionization suppression by as much as 70% has been achieved by Halfmann and co-workers in helium atoms [18,19] and up to 80% in xenon atoms [20].

When the trapping condition is fulfilled, coherent processes between the bound states become possible. The most prominent among these processes is coherent population transfer between the bound states. In particular, the counterintuitive arrangement of the light fields, as in STIRAP, allows for population transfer between the bound states through the continuum state, with little or even no ionization [21,22,23,24]. In experiments, population transfer efficiency of about 20% has been achieved [20,25]. Optimal population transfer occurs only if the trapping condition on the two-photon detuning is achieved, especially if it is satisfied during the entire evolution of the system [26]. Because the driving fields are time-dependent, the trapping condition becomes time-dependent too. This requirement can be met to some extent by techniques like pulse chirping [27,28,29] and non-ionizing Stark shifts [29,30]. Incoherent decay channels can be suppressed by coupling of additional states to the system [31]. Viewed from the opposite angle, LICS can also be used to increase ionization via STIRAP into a continuum [32].

The standard LICS problem, reviewed by Knight [33], describes a Raman type transition between two bound states with intermediate common continuum state. Further theoretical development expands the model by including multiple continua [34], while keeping two bound states. Three bound states coupled to a continuum have been studied too [35], and, in this system, two trapping conditions have been derived. An attempt to include a sub-level structure of the bound states was realized by switching to the Laplace domain [36,37], which has the limitation of specific excitation patterns. In addition, simply adding more bound states to the standard LICS model drastically increases the complexity and prevents the derivation of a trapping condition because, in order to find such, one has to solve a characteristic polynomial equation of growing order as the number of states increases.

Recently, continuum structures such as the Fano profiles have been used extensively in a variety of fields, including atoms and molecules, nano-plasmonics, femtosecond and attosecond physics, and analogues of it in classical optics [38]. Various advances have been made in exploring and using Fano profiles of autoionization and LICS in atoms and molecules [39,40,41,42,43,44,45,46]. Notable examples include Rydberg states [39], effects of particle statistics [43], and double continua [44], to mention just a few. In nanophotonics, Fano effects have been observed and exploited various systems, including plasmonic nanostructures [47,48,49,50,51,52,53,54,55,56,57], metamaterials [47,58,59], semiconductors [60,61,62], and photonic crystals [63,64]. Their unique properties are utilized in optical filtering, polarization selectors, sensing, lasers, modulators, and nonlinear optics. Fano resonances have been extensively used in attosecond dynamics [65,66,67,68,69,70]. Analogues of Fano effects have been proposed and demonstrated in classical optics [71,72,73,74]. In addition, Fano resonances appear frequently in absorption spectroscopy and photo-angular electron distributions [75,76,77].

Other experimental demonstrations of LICS revealed interesting effects about chemical reactions [46,78], ionization branching [79], and harmonic generation [80]. The recent developments in intense attosecond X-ray pulses allow the impulsive coupling of a few bound states through the continuum [81]. Interference patterns between ionization pathways [67,82], on the other hand, might experience similarity to LICS coupling. Further experiments in molecules [83] showed how vibrational states can influence interference patterns and emphasized the need for more complicated LICS models that account for molecular properties. Dynamical LICS-related effects like electron–hole pair dynamics[84] and the consequent charge migration [85] in molecules remain largely unstudied due to the lack of LICS models that account for multiple bound states and yet provide the trapping conditions under which the effects can be demonstrated.

In this paper, we treat the excitation dynamics of degenerate levels of ground and excited bound states, coupled to a common continuum, similar to the system in [86], only we allow for direct as well as cross couplings. In order to address problems with the growing complexity of a large number of bound states, we show how a multilevel LICS system can be reduced to independent sub-systems of smaller dimension, by a proper change of basis. We further derive the trapping condition for the population transfer between the bound states of the sub-systems. We explore the dynamics for different initial conditions and their Fano profiles, and we suggest new applications of LICS based on our findings.

This paper is organized as follows: in Section 2, we introduce the problem. Section 3 treats population trapping, different initializations of the system, and the associated Fano profiles. We conclude our presentation in Section 4.

## 2. Multilevel LICS System

The evolution of *i* ground bound states coupled to *j* excited bound states through a common continuum of energy $E_{\epsilon }$ with a pump and Stokes laser reads (ℏ=1),
(1)$$\begin{array}{cc} i\frac{d}{dt}a_{g_{1}}\left(t\right)=\omega_{g_{1}}a_{g_{1}}\left(t\right) & -\int_{0}^{\infty }\Omega_{g_{1}\epsilon ,\;p}\left(t\right)\cos\left(\omega_{p}t\right)a_{\epsilon }\left(t\right)d\epsilon ,\\ \vdots \\ i\frac{d}{dt}a_{g_{i}}\left(t\right)=\omega_{g_{i}}a_{g_{i}}\left(t\right) & -\int_{0}^{\infty }\Omega_{g_{i}\epsilon ,\;p}\left(t\right)\cos\left(\omega_{p}t\right)a_{\epsilon }\left(t\right)d\epsilon ,\\ i\frac{d}{dt}a_{e_{1}}\left(t\right)=\omega_{e_{1}}a_{e_{1}}\left(t\right) & -\int_{0}^{\infty }\Omega_{e_{1}\epsilon ,\;p}\left(t\right)\cos\left(\omega_{p}t\right)a_{\epsilon }\left(t\right)d\epsilon ,\\ \vdots \\ i\frac{d}{dt}a_{e_{j}}\left(t\right)=\omega_{e_{j}}a_{e_{j}}\left(t\right) & -\int_{0}^{\infty }\Omega_{e_{j}\epsilon ,\;p}\left(t\right)\cos\left(\omega_{p}t\right)a_{\epsilon }\left(t\right)d\epsilon ,\\ i\frac{d}{dt}a_{\epsilon }\left(t\right)=\omega_{\epsilon }a_{\epsilon }\left(t\right) & -\sum_{g_{i}}a_{g_{i}}\left(t\right)\Omega_{g_{i}\epsilon ,\;p}\left(t\right)\cos\left(\omega_{p}t\right)-\sum_{e_{j}}a_{e_{j}}\left(t\right)\Omega_{e_{j}\epsilon ,s}\left(t\right)\cos\left(\omega_{s}t\right), \end{array}$$
where $\Omega_{g_{i}\epsilon ,p}$ and $\Omega_{e_{j}\epsilon ,s}$ are the Rabi frequencies coupling the $|g_{i}\rangle$ and $|e_{j}\rangle$ bound states to the common continuum |ϵ〉. We have ignored incoherent ionization channels streaming from the cross laser coupling, e.g., the Stokes laser ionizing the ground bound states, since they can be suppressed [31].

We consider the simplest multilevel system that has more than one state in each ground and excited bound levels, namely two ground and two excited states as illustrated in Figure 1a. For the sake of simplicity, we shall ignore highly detuned transitions, continuum–continuum transitions, and we will assume the ground and excited levels to be degenerate.

In addition, if we introduce the operator notation
(2)$${\hat{\Omega }}_{k,l}a_{s}\left(t\right)=\int_{0}^{\infty }\Omega_{k\epsilon ,l}\left(t\right)\cos\left(\omega_{l}t\right)a_{k}\left(t\right)d\epsilon ,$$
where *k* runs over the bound states and *l* runs over the lasers, we can re-write the system in matrix form
(3)$$i\frac{d}{dt}A\left(t\right)=H_{Sch}\left(t\right)A\left(t\right),$$
driven by a Hamiltonian, which, in the Schrodinger picture, reads
(4)$$H_{Sch}=\left[\begin{array}{ccccc} \omega_{g_{1}} & 0 & 0 & 0 & -{\hat{\Omega }}_{g_{1},p}\\ 0 & \omega_{g_{2}} & 0 & 0 & -{\hat{\Omega }}_{g_{2},p}\\ 0 & 0 & \omega_{e_{1}} & 0 & -{\hat{\Omega }}_{e_{1},s}\\ 0 & 0 & 0 & \omega_{e_{2}} & -{\hat{\Omega }}_{e_{2},s}\\ -{\hat{\Omega }}_{g_{1},p} & -{\hat{\Omega }}_{g_{2},p} & -{\hat{\Omega }}_{e_{1},s} & -{\hat{\Omega }}_{e_{2},s} & \omega_{\epsilon } \end{array}\right].$$

By formally integrating the equation for the continuum’s amplitude (the last one of Equations (Equation 1)) and substituting back into the equations for the bound states (see [33]), we can eliminate the continuum and give the evolution of the system in an effective picture of interacting bound states as illustrated in Figure 1b. Further changing the phase by $a_{n}\left(t\right)\to c_{n}\left(t\right)=a_{n}\left(t\right)\exp\left(i\omega_{n}t\right)$ transforms the effective system, in matrix form, to
(5)$$i\frac{d}{dt}C\left(t\right)=H\left(t\right)C\left(t\right).$$

The Hamiltonian driving the reduced system is a non-Hermitian matrix given by
(6)$$H\left(t\right)=-\frac{1}{2}\left(H_{0}+iH_{1}\right),$$
with
(7a)$$\begin{array}{cc} H_{0} & =\left[\begin{array}{cccc} -2\delta S_{g} & q_{\mathrm{gg}}\Gamma_{g} & q_{\mathrm{eg}}\Gamma_{\mathrm{eg}} & q_{\mathrm{eg}}\Gamma_{\mathrm{eg}}\\ q_{\mathrm{gg}}\Gamma_{g} & -2\delta S_{g} & q_{\mathrm{eg}}\Gamma_{\mathrm{eg}} & q_{\mathrm{eg}}\Gamma_{\mathrm{eg}}\\ q_{\mathrm{eg}}\Gamma_{\mathrm{eg}} & q_{\mathrm{eg}}\Gamma_{\mathrm{eg}} & -2\left(\Delta +\delta S_{e}\right) & q_{\mathrm{ee}}\Gamma_{e}\\ q_{\mathrm{eg}}\Gamma_{\mathrm{eg}} & q_{\mathrm{eg}}\Gamma_{\mathrm{eg}} & q_{\mathrm{ee}}\Gamma_{e} & -2\left(\Delta +\delta S_{e}\right) \end{array}\right], \end{array}$$
(7b)$$\begin{array}{cc} H_{1} & =\left[\begin{array}{cccc} \Gamma_{g} & \Gamma_{g} & \Gamma_{\mathrm{eg}} & \Gamma_{\mathrm{eg}}\\ \Gamma_{g} & \Gamma_{g} & \Gamma_{\mathrm{eg}} & \Gamma_{\mathrm{eg}}\\ \Gamma_{\mathrm{eg}} & \Gamma_{\mathrm{eg}} & \Gamma_{e} & \Gamma_{e}\\ \Gamma_{\mathrm{eg}} & \Gamma_{\mathrm{eg}} & \Gamma_{e} & \Gamma_{e} \end{array}\right]. \end{array}$$

The diagonal elements in Equations (7) are defined as follows. First of all,
(8)$$\Delta =E_{e_{i}}-E_{g_{i}}+\omega_{s}-\omega_{p}$$
is the reduced two-photon detuning connecting the ground and excited states through the continuum. Since we assume degeneracy, Δ is the same for both bound levels. The ionization rate due to a single laser is given by
(9)$$\Gamma_{k}=\frac{1}{2}\pi {\left|\Omega_{k\epsilon ,l}\right|}^{2},$$
where *k* runs over the bound states *g* and *e*, and *l* runs over the lasers. The Stark shifts caused by the lasers are defined as
(10)$$\delta S_{k}=-P.V⨋d\epsilon \frac{{\left|\Omega_{k\epsilon ,l}\right|}^{2}}{4(E_{\epsilon }-E_{k}-\omega_{l})},$$
where $E_{k}$ is the energy of the respective bound state, $E_{\epsilon }$ is the energy of the continuum state, and P.V. is the principal value of the integral. The off-diagonal elements
(11)$$\Gamma_{ij}=\frac{1}{2}\pi \Omega_{i\epsilon ,a}\Omega_{j\epsilon ,b}=\sqrt{\Gamma_{i}\Gamma_{j}},\;i\neq j$$
give the two-photon coupling between the bound ground *i* and excited *j* states through the continuum state |ϵ〉 due to the interaction with both lasers.

The continuum affects the evolution of the system by the Fano parameters
(12)$$q_{ij}=\frac{P.V.⨋d\epsilon \frac{\Omega_{i\epsilon ,l}\Omega_{j\epsilon ,m}^{*}}{2(E_{\epsilon }-E_{k}-\omega_{l})}}{\Gamma_{ij}},$$
where $\omega_{l}$ is the frequency of each laser *l* which drives the respective bound → continuum transition, while *m* indicates the laser driving the continuum → bound transition. For example, l=p=m for |g〉→|ϵ〉→|g〉 transition, and l=p, m=s for |g〉→|ϵ〉→|e〉 transitions. Since we consider a degenerate system, we can distinguish between three different Fano parameters, namely (i) $q_{gg}$ for transitions linking $|g_{1}\rangle$ and $|g_{2}\rangle$ through the continuum, (ii) $g_{ee}$ for transitions linking $|e_{1}\rangle$ and $|e_{2}\rangle$, and (iii) transitions between bound states through the continuum $|g_{i}\rangle$ and $|e_{j}\rangle$.

The elimination of the continuum state reduces the problem to four inter-coupled bound states. Although this elimination simplifies the system, it does not amount to solving the problem. For example, if we try to derive a population trapping condition by imposing conditions on the characteristic polynomial of Equation (Equation 6), as in [33], we have to solve a fourth-order equation for the eigenvalues of the system. If we add more states to the system, the problem becomes algebraically unsolvable in this basis.

A way out of this problem comes from a similarity transformation streaming from the idea employed in [87],
(13)$$U=\left[\begin{array}{cc} R(\theta ) & 0\\ 0 & R(\theta ) \end{array}\right],$$
where the matrix R reads
(14)$$R=\left[\begin{array}{cc} \cos(\theta ) & \sin(\theta )\\ -\sin(\theta ) & \cos(\theta ) \end{array}\right].$$
Thus, we can reduce the system to independent sub-systems, and allow for a simpler derivation of the trapping condition by fixing the rotation angle at θ=π/4, and further applying the shift transformation
(15)$$P=\left[\begin{array}{cccc} 1 & 0 & 0 & 0\\ 0 & 0 & 1 & 0\\ 0 & 1 & 0 & 0\\ 0 & 0 & 0 & 1 \end{array}\right].$$
The transformed Hamiltonian then reads
(16)$$\begin{array}{cc} \tilde{H} & =PUHU^{†}P=\left[\begin{array}{cc} H_{b} & 0\\ 0 & H_{d} \end{array}\right], \end{array}$$
where the matrices $H_{b}$ and $H_{d}$ are given by
(17a)$$\begin{array}{cc} H_{b} & =\left[\begin{array}{cc} \delta S_{g}-\frac{1}{2}\left(q_{\mathrm{gg}}+2i\right)\Gamma_{g} & -\left(q_{\mathrm{eg}}+i\right)\sqrt{\Gamma_{e}\Gamma_{g}}\\ -\left(q_{\mathrm{eg}}+i\right)\sqrt{\Gamma_{e}\Gamma_{g}} & \Delta -\frac{1}{2}\left(q_{\mathrm{ee}}+2i\right)\Gamma_{e}+\delta S_{e} \end{array}\right], \end{array}$$
(17b)$$\begin{array}{cc} H_{d} & =\left[\begin{array}{cc} \frac{q_{\mathrm{gg}}\Gamma_{g}}{2}+\delta S_{g} & 0\\ 0 & \Delta +\frac{q_{\mathrm{ee}}\Gamma_{e}}{2}+\delta S_{e} \end{array}\right]. \end{array}$$

The combined transformation upon the state vector reads,
(18)$$PUC\left(t\right)=\frac{1}{\sqrt{2}}\left[\begin{array}{c} c_{g_{1}}+c_{g_{2}}\\ c_{e_{1}}+c_{e_{2}}\\ c_{g_{2}}-c_{g_{1}}\\ c_{e_{2}}-c_{e_{1}} \end{array}\right]=\left[\begin{array}{c} b_{g}\\ b_{e}\\ d_{g}\\ d_{e} \end{array}\right].$$
In the next section, we investigate the behavior of the reduced system, and derive the conditions for population trapping.

## 3. Excitation Probabilities and Fano Profiles

The benefit of the block-diagonalization of Equation (Equation 6) is that now $H_{b}$ and $H_{d}$ operate independently on the superposition states $\left\{b_{g},b_{e}\right\}$ and $\left\{d_{g},d_{e}\right\}$, respectively. This allows us to derive separate trapping conditions. In our current example, this is solely for $H_{b}$ since $H_{d}$ is composed of “dark” states, which do not participate in the excitation. The general procedure to find a trapping condition for any Hamiltonian is to solve its characteristic polynomial for λ, such that
(19)ImdetH−λI=0,
and then impose conditions on Δ such that, upon substitution of λ of Equation(Equation 19) in
(20)RedetH−λI=0,
the last equation will hold. This procedure now underlines the benefit of the transformation of the Hamiltonian to independent blocks, since now the determinant of the whole Hamiltonian is the products of the determinants of the individual blocks which are of smaller dimension. Alternatively, if the system has more states than the order of the characteristic polynomial equation which we can solve, a trapping condition can not be derived. Among the multiple solutions for Δ, which the above procedure gives, we have to pick the one which is (i) physically meaningful and (ii) minimizes the ionization of a specific block; for example, if we want to have coherent transport of population to the excited bound states, we have to aim at preserving the population of $H_{b}.$ We require to have a real eigenvalue of $H_{b}$, since real eigenvalues lead to imaginary exponents in the solution for the state vector, which indicates that the population is trapped within the bound states. If we initialize the system such that only the bright states participate in the interaction, we need to derive a trapping condition only for the bright Hamiltonian.

Thus, we find the trapping condition to be
(21)$$\Delta =\frac{1}{2}\left(\Gamma_{e}q_{\mathrm{ee}}-\Gamma_{g}q_{\mathrm{gg}}\right)+q_{\mathrm{eg}}\left(\Gamma_{g}-\Gamma_{e}\right)+\delta S_{g}-\delta S_{e}.$$
The main difference between the trapping condition of a standard two-level LICS and our model is the additional term deriving from the couplings of the same bound level through the continuum. In order to explore this difference, we turn to the solution of the system driven by $H_{b}.$ For the sake of simplicity, we look at continuous-wave (cw) excitation where all units have been normalized to a characteristic time scale appropriate for a specific system *T*. For cw radiation, it can be the time over which the lasers illuminate the system, but, alternatively, it can be any time to which we choose to normalize. In an experiment with cw lasers, illumination can continue until the sample is either fully ionized, or until the ionization has saturated; for that matter, *T* can be chosen with respect to any of these moments of time. If pulsed excitation is used, the time dependence for the couplings and the detuning has to be the same [26], which is not a particular difficulty for modern laser systems.

### 3.1. System Initialized in a Coherent Superposition of States

In the simplest scenario, we initialize the system in a coherent superposition of the ground states $\left[c_{g_{1}}\left(0\right)+c_{g_{2}}\left(0\right)\right]/\sqrt{2}=b_{g}\left(0\right)=1$. This initial condition ensures that the dark states will not be populated and the evolution of the system will be governed by
(22)$$i\frac{d}{dt}B=H_{b}B,$$
with $B={\left[b_{g},b_{e}\right]}^{T}$.

The general solution of Equation (Equation 22) is too cumbersome to be presented here even for cw excitation. A simplified solution can be generated if we substitute the trapping condition of Equation (Equation 21) back into the solution, which then reads
(23a)$$\begin{array}{cc} b_{g} & =\frac{\left[\Gamma_{e}+\Gamma_{g}e^{it\left(q_{\mathrm{eg}}+i\right)\left(\Gamma_{e}+\Gamma_{g}\right)}\right]e^{-\frac{1}{2}it\left(\Gamma_{g}\left(2q_{\mathrm{eg}}-q_{\mathrm{gg}}\right)+2\delta S_{g}\right)}}{\Gamma_{e}+\Gamma_{g}}, \end{array}$$
(23b)$$\begin{array}{cc} b_{e} & =\frac{\sqrt{\Gamma_{e}\Gamma_{g}}\left[e^{it\left(q_{\mathrm{eg}}+i\right)\left(\Gamma_{e}+\Gamma_{g}\right)}-1\right]e^{-\frac{1}{2}it\left(\Gamma_{g}\left(2q_{\mathrm{eg}}-q_{\mathrm{gg}}\right)+2\delta S_{g}\right)}}{\Gamma_{e}+\Gamma_{g}}. \end{array}$$

The evolution of the system under the trapping condition is shown in Figure 2.

Finally, we bring a brief discussion about the initialization of the system in any of the dark states. At first glance, nothing interesting can happen since no population transfer between the bound states occurs. However, due to destructive interference in the induced laser structure, also no ionization occurs. The only ionization that is allowed from such state has to come from an incoherent channel that is always present in a real experiment. This effect can not be observed when only two bound states are involved because the dark states are superpositions of bound states from the same level, thus the standard (two bound states) LICS system has none.

### 3.2. System Initialized in One of the Ground States

The multilevel LICS problem is quite sensitive to the initial condition. If we instead initialize the system in one of its ground states, say $c_{g_{1}}\left(0\right)=1$, we will have to account for one of the dark states, namely $d_{g}$. The evolution of the system is then governed by
(24)$$i\frac{d}{dt}\left[\begin{array}{c} B\\ d_{g} \end{array}\right]=\left[\begin{array}{cc} H_{b} & 0\\ 0 & \frac{q_{\mathrm{gg}}\Gamma_{g}}{2}+\delta S_{g} \end{array}\right]\left[\begin{array}{c} B\\ d_{g} \end{array}\right].$$
Due to the block-diagonal form of Equation (Equation 24), the solution for B remains unchanged besides a factor accounting for the new initial condition. The solution of Equation (Equation 24) with an imposed trapping condition and accounting for the dark state reads
(25a)$$\begin{array}{cc} b_{g} & =\frac{\left[\Gamma_{e}+\Gamma_{g}e^{it\left(q_{\mathrm{eg}}+i\right)\left(\Gamma_{e}+\Gamma_{g}\right)}\right]e^{-\frac{1}{2}it\left[\Gamma_{g}\left(2q_{\mathrm{eg}}-q_{\mathrm{gg}}\right)+2\delta S_{g}\right]}}{\sqrt{2}\left(\Gamma_{e}+\Gamma_{g}\right)}, \end{array}$$
(25b)$$\begin{array}{cc} b_{e} & =\frac{\sqrt{\Gamma_{e}\Gamma_{g}}\left[e^{it\left(q_{\mathrm{eg}}+i\right)\left(\Gamma_{e}+\Gamma_{g}\right)}-1\right]e^{-\frac{1}{2}it\left[\Gamma_{g}\left(2q_{\mathrm{eg}}-q_{\mathrm{gg}}\right)+2\delta S_{g}\right]}}{\sqrt{2}\left(\Gamma_{e}+\Gamma_{g}\right)}, \end{array}$$
(25c)$$\begin{array}{cc} d_{g} & =-\frac{e^{-\frac{1}{2}it\left(2\delta S_{g}+\Gamma_{g}q_{\mathrm{gg}}\right)}}{\sqrt{2}}. \end{array}$$
The population evolution is depicted in Figure 3. We note the difference in the ionization as well as the diminishing excitation of the bright states. This is the direct consequence of the initial condition. With the initial condition $c_{g_{1}}\left(0\right)=1$, the ionization
(26)$$I=1-{\left|b_{g}\right|}^{2}-{\left|b_{e}\right|}^{2}-{\left|d_{g}\right|}^{2}$$
decreases, since the dark state tends to preserve half of the population among $c_{g_{1}}$ and $c_{g_{2}}.$ Consequently, less population can pass through the continuum to the excited bound states. This behavior outlines the importance of initializing the system at $b_{g}\left(0\right)=1,$ since, in that case, the term $|d_{g}{|}^{2}$ in Equation (Equation 26) vanishes.

### 3.3. Fano Profile

Finally, we point out significant differences between the Fano ionization profiles of our four-level model and the two-level model of [33], whose evolution is driven by
(27)$$H_{2lvl}\left[\begin{array}{cc} \delta S_{g}-\frac{i\Gamma_{g}}{2} & -\frac{1}{2}\left(q_{\mathrm{eg}}+i\right)\sqrt{\Gamma_{e}\Gamma_{g}}\\ -\frac{1}{2}\left(q_{\mathrm{eg}}+i\right)\sqrt{\Gamma_{e}\Gamma_{g}} & \Delta -\frac{i\Gamma_{e}}{2}+\delta S_{e} \end{array}\right].$$
The structure of $H_{b}$ and $H_{2lvl}$ is very similar. The excitation differs by a factor of $\frac{1}{2}$, and the diagonal elements are effectively shifted, so naturally one can expect the same Fano profile, whose minimum is also shifted. In order to avoid confusion, we note that the states upon which the two Hamiltonians act are different. The ionization of the four level Hamiltonian is given by Equation (Equation 26), whenever the system is initialized in the ground bound states, while the ionization of $H_{2lvl}$ is given by
(28)$$I=1-{\left|c_{g}\right|}^{2}-{\left|c_{e}\right|}^{2},$$
since the system is composed of only two bound states, namely |g〉 and |e〉.

In Figure 4, we show the ionization profiles for different models and different initial conditions. As we see in the figure, the profile of the two-level system is similar to the the profile of the four-level system initialized in a bright state. This is not the case for a system initialized in the $|g_{1}\rangle$ ground states. Looking at the ionization when both the two-state and four-state models are initialized in state $|g_{1}\rangle$, we find that the minimum of the ionization of the two-state model can correspond to a significant ionization of the four-state model. This mismatch shows that approximations ignoring the presence of nearby laying states, streaming for example from magnetic quantum number, do not predict the correct Fano resonance. Due to the dark state, the ionization of the system can not exceed $\frac{1}{2},$ since it keeps half of the population inside the bound states. This last feature is quite significant and can be harnessed in a few useful ways. For example, if we want to break a chemical bond which is surrounded by multiple states, it is best if we first create a coherent superposition and then ionize at a maximum of the Fano profile. Alternatively, by comparing Fano profiles, we can judge about the structure of the system, since the more dark states are involved, the smaller the ionization, as each dark state will tend to keep more population bounded.

Finally, we want to point out that a fulfilled trapping condition does not mean vanishing ionization but rather a minimum. It ensures that one of the eigenvalues of the Hamiltonian will be real, but not all. Thus, a decay channel is open through the states with complex eigenvalues.

### 3.4. Non-Degeneracy

Hitherto, we have assumed that the states in each level are degenerate. At first glance, it appears as a strong condition on the system not only because real systems are non-degenerate, but it also equalizes the strength of all bound–continuum–bound transitions, as well as the Stark shifts and the Fano parameters. However, such differences will be of the order of the energy shifts among the states in the bound levels and are also quite small compared to the ionization couplings $\Gamma_{ij}$. In order to estimate under what circumstances we can treat the system as degenerate, we investigate numerically the evolution of the non-degenerate system driven by the Hamiltonian
(29)$$H_{nd}\left(t\right)=\left[\begin{array}{cc} \Delta_{g} & \Gamma \\ \Gamma & \Delta_{e} \end{array}\right],$$
where Γ are the coupling matrices of the degenerate Hamiltonian of Equation (7) and the modified ground and excited blocks lye on the diagonal,
(30a)$$\Delta_{g}=\left[\begin{array}{cc} \delta S_{g}-\frac{i\Gamma_{g}}{2} & -\frac{1}{2}\left(q_{\mathrm{gg}}+i\right)\Gamma_{g}\\ -\frac{1}{2}\left(q_{\mathrm{gg}}+i\right)\Gamma_{g} & \delta S_{g}-\frac{i\Gamma_{g}}{2}+\delta_{g} \end{array}\right],$$
(30b)$$\Delta_{e}=\left[\begin{array}{cc} \Delta -\frac{i\Gamma_{e}}{2}+\delta S_{e} & -\frac{1}{2}\left(q_{\mathrm{ee}}+i\right)\Gamma_{e}\\ -\frac{1}{2}\left(q_{\mathrm{ee}}+i\right)\Gamma_{e} & \Delta +\delta S_{e}-\frac{i\Gamma_{e}}{2}+\delta_{e} \end{array}\right],$$
which account for the non-degeneracy by the energy shifts between the bound states $\delta_{g}$ and $\delta_{e}.$

The effect of the non-degeneracy over time, depicted in Figure 5a, is to gradually increase the ionization and deplete the bounded states, thus to destroy the trapping. The effect of the degeneracy diminishes as the ratio of $\delta_{k}/\Gamma_{ij},$ which also regulates the width of the Fano profile (see Figure 5b). For a large ratio, the additional couplings proportional to the energy shifts become significant and destroy the bright and dark states picture. As evident (green line of Figure 5b), these couplings can allow a higher amount of the population to be ionized when the trapping condition is not met. The minima of the Fano profiles, however, do not shift significantly, although the width changes at larger times (blue line). Overall, for reasonable time scales and large enough ionization couplings, the system can be treated as degenerate. We note, however, that, due to the nonlinear dependence of the system’s response to the ionization couplings, such calculation for the validity of the degenerate treatment should always be carried out in order to determine the ionization strength and time window over which it is valid, as well as the error in the probabilities due to the approximation of degeneracy.

## 4. Conclusions

In this paper, we explored the multistate LICS process consisting of two ground and two excited degenerate bound states coupled through a common continuum. We reduced the dynamics of the system to a block-diagonal form by a rotation, mapping the evolution to bright and dark sub-systems. This reduction of the complexity allowed us to derive separate analytical trapping conditions for the sub-systems, in our case only for the bright Hamiltonian of Equation (Equation 22) since the dark states remain uncoupled. The assumption of degeneracy holds well as long as the ionization strength is large enough compared to the energy shifts between the bound states. Furthermore, we showed that the Fano profiles strongly depend on the initialization of the system. Initially populated bright state reproduces a standard ionization profile, while initialization in one of the ground states in the original basis sets an upper bound of the ionization of $\frac{1}{2}$ because half of the population is trapped in the dark state. The latter feature of the system can be an indicator of the number of states involved in the interaction and thus probes the structure of the system. Although our model does not yet incorporate electron–hole dynamics, it paves the way towards such extension, since it demonstrates what kind of effects can be expected, for example, shifting the ionization minimum or decoupling of the bound states. Another important application of the multistate LICS can be the generation of coherent superpositions of Rydberg ions. Naturally, the Rydberg states are lying close to the continuum and can serve as the excited states in our model. Thus, a sample of Rydberg atoms initially prepared in a coherent superposition of ground states can be mapped through the continuum to an excited state. In addition, the fact that the different initial conditions generate strikingly different evolution can be harnessed in the field of chiral chromatography. Cyclic optical excitation can initialize the enantiomers in the $b_{g}$ and $d_{g}$ states, respectively, which in turn will give different ionization profiles of the enantiomers that can allow for their spatial separation. Such application can not be generated by the conventional LICS. 

## Figures and Tables

**Figure 1 entropy-23-00891-f001:**
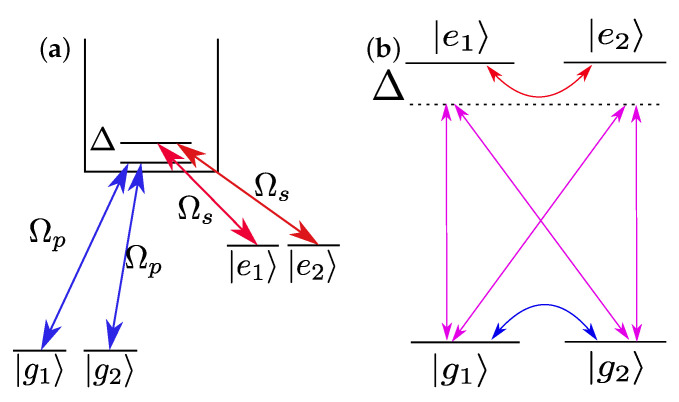
(Color online) Multilevel LICS coupling scheme. (**a**) Four bound states coupled to common continuum. (**b**) The effectively reduced system of inter-coupled bound states, by elimination of the continuum.

**Figure 2 entropy-23-00891-f002:**
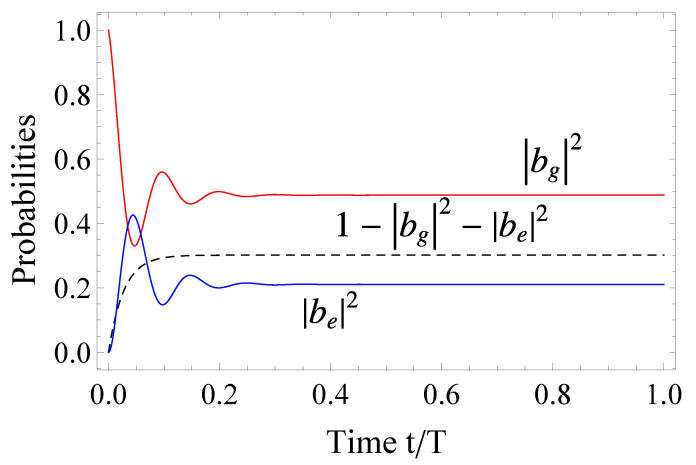
(Color online) Probability amplitudes versus normalized time t/T for system initialized in the "bright" ground state. The red and blue lines give the ground and excited bright states, respectively, while the dashed black line depicts the ionization probability. The excitation parameters are set to $\delta S_{g}=0.5T^{-1}$, $\delta S_{e}=0.6T^{-1}$, $\Gamma_{g}=5.5T^{-1},$$\Gamma_{e}=12.74T^{-1},$$q_{gg}=2.3$, $q_{eg}=3.4$, $q_{ee}=5$.

**Figure 3 entropy-23-00891-f003:**
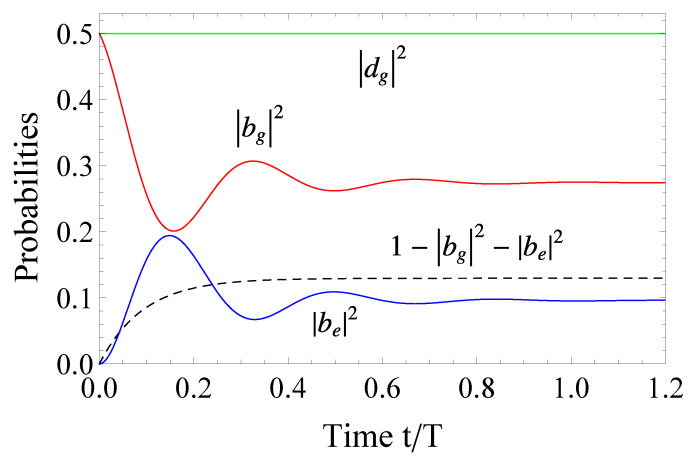
(Color online) The same as Figure 2 but for the system initially in state $|g_{1}\rangle$. The dashed black line depicts the ionization probability, while the green line gives the population in the dark state. The excitation parameters are set to $\delta S_{g}=0.5T^{-1}$, $\delta S_{e}=0.6T^{-1}$, $\Gamma_{g}=5.5T^{-1},$$\Gamma_{e}=12.74T^{-1},$ $q_{gg}=2.3$, $q_{eg}=3.4$, $q_{ee}=5$.

**Figure 4 entropy-23-00891-f004:**
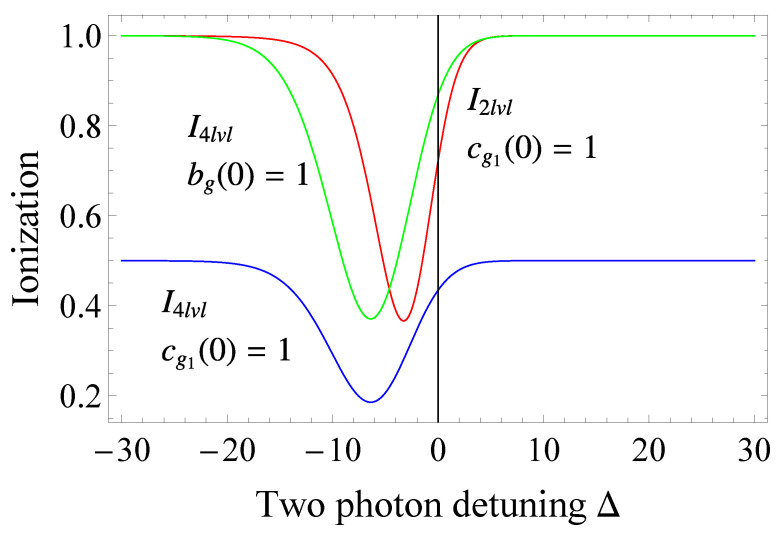
(Color online) Ionization versus two photon detuning for different system configurations. The red line shows the ionization for the standard two-state LICS [33]. The blue and green lines show the ionization for the four-state model of Equation (Equation 26) with the population initially respectively in state $|g_{1}\rangle$ and the bright state $|b_{g}\rangle$. The excitation parameters are set to $\delta S_{g}=0.5T^{-1}$, $\delta S_{e}=0.6T^{-1}$, $\Gamma_{g}=5.5T^{-1},$$\Gamma_{e}=12.74T^{-1},$$q_{gg}=2.3$, $q_{eg}=3.4$, $q_{ee}=5$, t/T=6.

**Figure 5 entropy-23-00891-f005:**
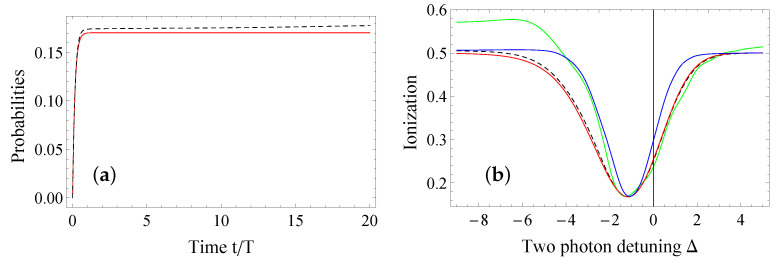
(Color online) (**a**) Ionization as a function of the normalized time for the degenerate Hamiltonian (red solid line) of Equation (7) and the non-degenerate (black dashed line) of Equation (Equation 29) for the system initialized in $c_{g_{1}}\left(0\right)=1$. The excitation parameters are set to $\Gamma_{g}=1.08T^{-1},\;\Gamma_{e}=2.09T^{-1},\;\delta S_{g}=0.33T^{-1},\;\delta S_{e}=0.26T^{-1},\;\delta_{g}=\delta_{e}=0.2T^{-1},\;q_{gg}=2.3,\;q_{eg}=2.4,\;q_{ee}=2.5$. (**b**) Fano profiles for the degenerate and non-degenerate systems. The red line is the degenerate profile, while the blacked dashed line ($\delta =0.02T^{-1}$), the green ($\delta =0.2T^{-1}$) and the blue ($\delta =0.02T^{-1}$) solid lines are the non-degenerate profiles calculated at t/T=10, except the blue line, which is at t/T=20..

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
