# Peer review of "Multilevel Laser Induced Continuum Structure"

_entropy, 2021, doi:10.3390/e23070891_

Round 1

Reviewer 1 Report

The authors consider multi-state generalization of laser-induced-continuum-structure(LICS) which couples two bound levels, each composed of two degenerate states through a common continuum state. The authors analyzed the trapping condition (ionization suppression)  in the system. Also, they demonstrate dependence of the LICS on the initial conditions. Comparison between the four-level model and the two-level model results is provided.

There are a few concerns about clarity of the Eq. (2), (3)  and sub-scripts, like \Omega_{i, \epsilon, l}  in Eq. (3) and \Omega_{k c_{\epsilon}, l}  in Eq. (8). Sure it's just notation, but there is no explanation in the manuscript what is \Omega_{i,  \epsilon, l}.

The term "The operator Rabi frequencies" is too inventive and it is confusing. 

Also, I believe that the elimination procedure of the continuum should be clarified. A clear description of the "continuum states" should be provided.

The presented results are very interesting and manuscript might generate interesting discussions in the quantum physics  and control community.

The manuscript could be published after adjustments indicated above.

Author Response

 We thank the referee for reading and reviewing our manuscript. 
Our first referee has no objections on the results in the manuscript but he/she is rather concerned with the presentation of the derivation of the starting Hamiltonian and some minor notational problems.

In order to address the referee's points we have carefully revised our manuscript. We have colored the changes in the text in red for easy tracking.

--------------------------------------------------------------
Reply to the First Referee -- entropy-1274599
--------------------------------------------------------------

1. Referee: There are a few concerns about clarity of the Eq. (2), (3)  and sub-scripts, like \Omega_{i, \epsilon, l}  in Eq. (3) and \Omega_{k c_{\epsilon}, l} in Eq. (8). Sure it's just notation, but there is no explanation in the manuscript what is \Omega_{i,  \epsilon, l}. The term "The operator Rabi frequencies" is too inventive and it is confusing. 

Reply: We agree with the referee and we changed the notation.

Change: Added definitions and changed notations.

2. Referee: Also, I believe that the elimination procedure of the continuum should be clarified. A clear description of the "continuum states" should be provided.

Reply: The procedure we use for the elimination of the continuum is well established and has been followed in a number of papers and for that matter we restrained ourselves from a lengthy discussion. However we followed the referee's recommendation and added some clarifications into the technical aspect of the elimination. 

Change: Explained the elimination procedure.

Reviewer 2 Report

The manuscript „Multilevel Laser Induced Continuum Structure“ given by Kaloyan Zlatanov and Nikolay Vitanov presents a theoretical investigation of the atomic population transfer from the ground to the excited states though the continuum in the presence of ionization.

The authors extend the two-bound-level model to four-bound level and show the importance of taking into account the additional levels, for example, for the ionization profiles.

The work presents an interesting study, however, the following questions/ comments appear:

  • P and Nu in Eq. 10 are not defined
  • The authors use several transformations (a rotation with an angle Pi/4 and a shift transformation) to diagonalize Hamiltonian. Why are directly these transformations chosen ? Is this diagonalization unique?
  • In p.5 the authors write ‘In order to find the trapping condition, we require to have a real eigenvalue of Hb’. It is better to specify here that the real eigenvalues lead to imaginary exponents in the solution for the state vector, and as a consequence, to the trapping (rather than ionization). Otherwise, once can choose the solution in terms of real exponents, and the real eigenvalues will lead to the ionization.
  • It is not clear how the trapping condition Eq.18 was obtained, since the characteristic equation has several real solutions
  • Why is the coupling strength of the excited state Gamma_e about twice bigger than the coupling strength of the ground state Gamma_g ?
  • What is the meaning of time period T for CW laser ?
  • In p.6 the authors write ‘Finally we bring a brief discussion about the initialisation of the system in a dark state. At first glance nothing interesting can happen since no population transfer between the bound states occurs.’ It is not clear which initial condition is discussed here: d_g(0)=1 or d_e(0)=1 ? Moreover, the following sentence should be rewritten in a more clear way (mentioning destructive interference, etc): ‘However due to coherent interference in the induced laser structure also no ionisation occurs’  The authors continue: ‘This effect can not be observed when only two bound states are involved’.  Why  this effect can not be observed when only two bound states are involved?
  • 4 includes I_4lvl. Does this I_4lvl exactly coincide with the Eq. 23 ? It would be better to give also the expression for the I_2lvl (at least in the Appendix)
  • This is misleading that in Fig.4 the curves for I_4lvl and I_2lvl are similar for different initial conditions (b_g(0)=1 and c_g1(0)=1). As far as I understood this is because in the 2-level model c_g1(0)= b_g(0). But this is better to specify.
  • The authors show that accounting for additional two levels is important for ionization profiles. Is there a set of parameters for which one can neglect two additional levels and consider a simple two-bound-level model ? How will adding two more levels (considering six-bound-level model) modify the results?

Author Response

We thank the referee for reading and reviewing our manuscript. 
Our second referee points out some very important questions regarding the properties of the system and 
the significance of our model in relation to ionization physics.

In order to address the referee's points we have carefully revised our manuscript. We have colored the changes in the text in red for easy tracking.

--------------------------------------------------------------
Reply to the Second Referee -- entropy-1274599
--------------------------------------------------------------

1.Referee: P and Nu in Eq. 10 are not defined

Reply: We fixed the definitions.

2.Referee: The authors use several transformations (a rotation with an angle Pi/4 and a shift transformation) to diagonalize Hamiltonian. 
Why are directly these transformations chosen ? Is this diagonalization unique?

Reply: The complications with multiple states in this work derive from the bound->continuum->bound transitions from the same bound level. In the absence of such transitions the system can be reduced to independent sub-systems for example by a Morris-Shore transformation. These transitions, however, can not be ignored, but a way to treat them is to diagonalize the blocks lying on the diagonal. The diagonalization is by no means unique. In general, diagonalization of complex matrices is a more complicated task than the diagonalization of real matrices, with issues streaming from complex roots, for example, if the diagonalization is to be carried by transformation matrices composed of eigenvectors. In order to avoid such complications we looked for alternative transformations and as a starting point we employed the idea of Householder reflection as in Phys. Rev. A 77, 033404 (2008). The two-by-two case seems trivial, but once more states are added the transformation is not trivial at all. This issue is well outside the scope of the present paper.

3. Referee: In p.5 the authors write ‘In order to find the trapping condition, we require to have a real eigenvalue of Hb’. It is better to specify here that the real eigenvalues lead to imaginary exponents in the solution for the state vector, and as a consequence, to the trapping (rather than ionization). Otherwise, once can choose the solution in terms of real exponents, and the real eigenvalues will lead to the ionization. It is not clear how the trapping condition Eq.18 was obtained, since the characteristic equation has several real solutions

Reply: We have updated the paragraph and we added explanations for the derivation of the trapping condition and how to choose it among the roots of the  characteristic polynomial.

4. Referee: Why is the coupling strength of the excited state Gamma_e about twice bigger than the coupling strength of the ground state Gamma_g ?

Reply: Within our current model a fundamental reason for this is not obvious/present. However we ignore here the cross laser ionization, the pump laser ionizing the excited states and the Stokes laser ionizing 
the ground bound states and other multiphoton processes which are characteristic of specific system. In a real experiments, for example, PHYSICAL REVIEW A 66, 013406 (2002), PHYSICAL REVIEW A 58, 1 (1998)
usually the Stokes laser is about a factor of 10 times stronger, where the ratio between the laser strength is dictated by signal optimization and suppresion of incoherent ionization channels, 
hence our choice for the laser strength in the simulation.

5. Referee: What is the meaning of time period T for CW laser ?

Reply: We fixed the expression. T is a characteristic time scale which makes the units dimensionless and can be adjusted in accordance to experiment.

6. Referee: In p.6 the authors write ‘Finally we bring a brief discussion about the initialisation of the system in a dark state. At first glance nothing interesting can happen since no population transfer between the bound states occurs.’ It is not clear which initial condition is discussed here: d_g(0)=1 or d_e(0)=1 ? Moreover, the following sentence should be rewritten in a more clear way (mentioning destructive interference, etc): ‘However due to coherent interference in the induced laser structure also no ionisation occurs’  The authors continue: ‘This effect can not be observed when only two bound states are involved’.  Why  this effect can not be observed when only two bound states are involved?

Reply: We thank for this question since it underlines an important difference between the standard LICS and this work. We address any of the dark states present. A system initialized in any of the two dark states will remain in it without being ionized. Of course, this is not the experimental reality where incoherent ionization channels are present, but if they are suppressed the overall ionization from the dark states will be minimized, and due only to incoherent channels. This effect can not be observed when only two bound states are involved because the dark states are superpositions of bound states from the same level, thus the standard (two bound states) LICS system has none. 

7. Referee: 4 includes I_4lvl. Does this I_4lvl exactly coincide with the Eq. 23 ? It would be better to give also the expression for the I_2lvl (at least in the Appendix) This is misleading that in Fig.4 the curves for I_4lvl and I_2lvl are similar for different initial conditions (b_g(0)=1 and c_g1(0)=1). As far as I understood this is because in the 2-level model c_g1(0)= b_g(0). But this is better to specify.

The authors show that accounting for additional two levels is important for ionization profiles. Is there a set of parameters for which one can neglect two additional levels and consider a simple two-bound-level model ?

Reply: Yes, I_4lvl is given by Eq.(23) (Eq.(26) in the new manuscript), we added some clarification and we gave the ionization for the standard LICS model explicitly. The bright states are superpositions of the states in the same level, and therefore although the profiles for I_4lvl initialized at b_g(0)=1 look similar to the I_2lvl initialized at c_g(0)=1 it is incorrect to compare them. The similarity derives from the fact that they both have similar structure of the Hamiltonians that drive the systems. However, a more reasonable comparison is of I_4lvl initialized at c_g1(0)=1 and I_2lvl, since it demonstrates how the Fano profiles will look if we do not account for the additional bound states participating in the interaction. Under no circumstances/parameters the standard LICS can be used as an approximation, also this is what was reported in J. Phys. B: At. Mol. Opt. Phys. 32 (1999) 4485–4493, where LICS was observed in NO molecules.

8. Referee: How will adding two more levels (considering six-bound-level model) modify the results?

Adding more states changes the ionization profile and shifts the ionization extremum. Strong trapping and coherent population transfer to the bound states can be achieved only if the system is initialized in a superposition of states, under the trapping condition. Away from the trapping condition, and when the system is not in a superposition we see from Fig. 4 that the dark states will be active and they tend to "pull" population, thus lowering the ionization probability. In our model the maximum ionization probability for c_g1(0)=1 is 50%. Adding more states will make it drop as 1/N (N=number of states in a level). It is important to underline that this is true only if no cross laser ionization is present, that is only if the cross laser ionization is compensated as in Phys. Rev. A (1998), 57, 462.

Reviewer 3 Report

The manuscript “Multilevel Laser Induced Continuum Structure” by K. Zlatanov  and N. Vitanov belongs to the broad field laser induced structure in the continuum with different level schemes (lamda-scheme, m- scheme, w-scheme etc). The field started developing in the late 80th with researches by P.L. Knight, M.A. Lauder and B.J. Dalton and the author of the manuscript is one of the pioneers of these researches.

The authors are well aware about current background of the field and presented an essential introduction.

The applied method is traditional and well established for the field and based on effective Hamiltonian matrix. The results are not breakthrough, but they are substantial, clear and discussed in appropriate way.

I believe that the manuscript deserved to be published in “Entropy “.

Author Response

We thank the referees for reading and reviewing our manuscript. 

Our third referee is very positive in his/her assessment of our manuscript and recommends 
publication in its present form.

In order to address the referee's points we have carefully revised our manuscript. We have colored the changes in the text in red for easy tracking.

Round 2

Reviewer 2 Report

I think the manuscript has a nice form and I would be glad to 
 recommend it for publication if the following minor questions would 
 be resolved:

 1) this would be better to cite Phys. Rev. A 77, 033404 (2008) (as 
 in the reply) when the authors discuss the diagonalization of the 
Hamiltonian

 2) it is still not completely clear what the characteristic time is. 
Is it the range of ps or ns ?
 Probably, when one has a pulsed laser, the characteristic time is 
 connected with a pulse duration. What time scale should be assumed 
 in the case of a CW laser?

 3)In p.8 authors write 'we will have to account for one of the dark 
 states, namely dg'. Does it matter that we chose dg, not de ?

Author Response

We would like to resubmit our updated manuscript which is under consideration in Entropy. Our second referee has three points, which require 
further clarification.

In order to address the referee's points we have carefully revised our manuscript. We have colored the changes in the text in red for easy tracking.    

We hope that our clarifications and amendments correctly resolve our referee's concerns and the Editor will find our revised manuscript suitable for 
publication in the special issue of Entropy.

Sincerely,

K. N. Zlatanov and N. V. Vitanov

--------------------------------------------------------------
Reply to the Second Referee -- entropy-1274599
--------------------------------------------------------------

 1.Referee: this would be better to cite Phys. Rev. A 77, 033404 (2008) (as 
 in the reply) when the authors discuss the diagonalization of the 
Hamiltonian

 Reply: We cite the paper in the new manuscript.

 2.Referee: it is still not completely clear what the characteristic time is. 
 Is it the range of ps or ns ?
 Probably, when one has a pulsed laser, the characteristic time is 
 connected with a pulse duration. What time scale should be assumed 
 in the case of a CW laser?

 Reply: When pulsed lasers are used T will be the pulse duration. For cw radiation it can be the time over which the lasers illuminate the system
 but alternatively it can be any time to which we choose to normalize. In an experiment with cw lasers, illumination can continue until the sample 
 is either fully ionized, or until the ionization has saturated, for that matter T can be chosen with respect to any of these moments of time. 

 3.Referee: In p.8 authors write 'we will have to account for one of the dark 
 states, namely dg'. Does it matter that we chose dg, not de ?

 Reply: Yes it does, we account for d_g because it is a superposition of the ground states(c_g2-c_g1) and when we initialize the system in c_gi, the d_g state
 is active, it has initially accquired population that can not be transfered to any other state(by coherent channels). 
 Alternatively if we initialize the system in any of the excited bound states c_ei then we would have used d_e=(c_e2-c_e1).